# Tropical peat composition may provide a negative feedback on fire occurrence and severity

Alastair J. Crawford [1,2,15] ✉, Claire M. Belcher[1,15], Stacey New [3], Angela Gallego-Sala [4], Graeme T. Swindles [5,6], Susan Page[7], Tatiana A. Blyakharchuk [8,9], Hinsby Cadillo-Quiroz [10], Dan J. Charman [4], Mariusz Gałka[11], Paul D. M. Hughes[12], Outi Lähteenoja [16], Dmitri Mauquoy [13], Thomas P. Roland [4] & Minna Väliranta [14]

Loss of peat through increased burning will have major impacts on the global carbon cycle. In a normal hydrological state, the risk of fire propagation is largely controlled by peat bulk density and moisture content. However, where humans have interfered with the moisture status of peat either via drainage, or indirectly via climate change, we hypothesise that its botanical composition will become important to flammability, such that peats from different latitudes might have different compositionally-driven susceptibility to ignition. We use pyrolysis combustion flow calorimetry to determine the temperature of maximum thermal decomposition ($T_{max}$) of peats from different latitudes, and couple this to a botanical composition analysis. We find that tropical peat has higher $T_{max}$ than other regions, likely on account of its higher wood content which appears to convey a greater resistance to ignition. This resistance also increases with depth, which means that loss of surface peat in tropical regions may lead to a reduction in the subsequent ignitability of deeper peat layers as they are exposed, potentially resulting in a negative feedback on increased fire occurrence and severity.

Peat, unlike vegetation, burns almost exclusively by smouldering (non-flaming) combustion[1–3], resulting in greater emissions of CO and $CH_4$ and having a higher climate forcing potential[4], as well as greater emissions of toxins and particulates[5]. Despite covering 2–3% of the Earth's land surface[5,6], peatlands store around a quarter to a third of global soil carbon[5,7], probably exceeding the carbon content of global vegetation[5,6], and equalling or exceeding that of the pre-industrial atmosphere[8,9]. Increased burning of peatlands due to climate warming[5,10] and anthropogenic peatland degradation[6,11,12] thus risks substantial impacts on the carbon cycle and climate system[13,14], as well as other harmful environmental impacts[15,16].

[1]wildFIRE Lab, Hatherly Laboratories, University of Exeter, Exeter, UK. [2]School of Environment, Earth and Ecosystem Sciences, The Open University, Milton Keynes, UK. [3]Met Office, Exeter, UK. [4]Geography, Faculty of Environment, Science and Economy, University of Exeter, Exeter, UK. [5]Geography, School of Natural and Built Environment, Queen's University Belfast, Belfast, UK. [6]Ottawa-Carleton Geoscience Centre and Department of Earth Sciences, Carleton University, Ottawa, ON, Canada. [7]School of Geography, Geology and the Environment, University of Leicester, Leicester, UK. [8]Tomsk State University, Tomsk, Russia. [9]Institute of Monitoring of Climatic and Ecological Systems SB RAS, Tomsk, Russia. [10]School of Life Sciences and Biodesign Institute, Arizona State University, Tempe, AZ, USA. [11]Faculty of Biology and Environmental Protection, Department of Biogeography, Paleoecology and Nature Conservation, University of Lodz, Łódź, Poland. [12]Palaeoecology Laboratory, Department of Geography, University of Southampton, Southampton, UK. [13]School of Geosciences, University of Aberdeen, Aberdeen, UK. [14]Ecosystems and Environment Research Programme, University of Helsinki, Helsinki, Finland. [15]These authors contributed equally: Alastair J. Crawford, Claire M. Belcher. [16]Unaffiliated: Outi Lähteenoja. ✉e-mail: a.j.crawford2@exeter.ac.uk

Despite the short-term climate forcing associated with $CH_4$ production, peat accumulation has a net cooling effect on the global climate in the long term through carbon sequestration[5,9,17]. Throughout the Holocene, peatlands have acted as a long-term carbon sink[18], but are now increasingly switching from sink to source, due to changes in climate, land use, and fire regimes[6]. Even though pristine peatlands globally may increase their carbon accumulation potential with climatic warming during this century, any such increase will weaken from c.2100 when enhanced decomposition may exceed enhanced photosynthesis[8]. The response of peatlands to warming is, however, latitude-dependent and intact tropical peatlands are expected to experience a decrease in carbon accumulation rates in the future due to increased rates of respiration[8]. However, changes in the balance between productivity and respiration are likely to be negligible in comparison with reductions in the area of intact peatlands, especially in the tropics, where they are subject to extensive deforestation and drainage for agriculture[6,19]. This is of considerable concern because the stability of peatlands is highly dependent on hydrological conditions, and exposure of the peat to oxygenation by drawdown of the water table, either artificially or as a result of drought, results in peat loss, either by peat mineralisation (decomposition) or by burning[6,20]. Under peat-forming conditions, high moisture content prevents ignition. Thus undisturbed peatlands are largely free of fire in the tropics[5,21], while high-latitude peatlands burn with limited severity under natural conditions[5,22–25]. However, lowering the water table, for example, using agricultural drainage schemes, exposes flammable (i.e. non-saturated) peat and this is known to be a strong driver of increased fire occurrence in tropical peatlands[6,21,26], and is associated with increased burn severity in high latitude peatlands[27] where drying due to climate change may also lead to vegetation changes followed by progressive peat loss through repeated burning at higher fire frequencies[28]. Therefore, carbon release from peat fires represents an important component of the human-altered carbon cycle[29,30].

The smouldering combustion of peat fires[1–3,31] is markedly different in behaviour compared with aboveground vegetation fires, which are dominated by flaming combustion. The high porosity of peat allows ingress of oxygen and in situ oxidation of the fuel allowing smouldering combustion. When peat burns the fires are slow-moving, with spread rates of $1–10\,cm\,h^{-1}$[32,33], and of low temperature, with typical peak temperatures of 500–700 °C[32]. However, the low thermal conductivity of peat minimises heat loss, so that despite the relatively low temperature, combustion is highly persistent[1,32]. This enables peat fires to cover extensive areas and penetrate deep into the ground[1], burning for many months or even years[1]. The propagation of smouldering fires in peat is largely controlled by its bulk density and moisture content[34,35]. However, large peat fires most often occur in areas that have been extensively drained for land use[6,21]. For example, degraded tropical peat has been found to be as dry as 20% moisture[36], and with moisture content no longer high enough to suppress burning, the controls on flammability may be shifted toward variations in peat composition. However relatively little is known about the influence of the botanical constituents within peat on ignition[37].

Peat combustion is normally initiated by the heat flux from a flaming vegetation fire, which in tropical regions is often anthropogenic[21,38,39]. Although flaming and smouldering are distinct—smouldering is a heterogenous reaction of solid fuel with an oxidiser whilst flaming is a homogeneous reaction of gaseous fuel with an oxidiser—both fire types begin with pyrolysis[36], which is the thermal decomposition of materials at elevated temperature. The thermal resistance of peat to the energy flux from surface fires is important in understanding the ignitability of peat. The thermal degradation properties of peat have been shown to vary considerably with both botanical composition and elemental composition[37]. Therefore, differences in peat composition, for example between tropical and boreal environments[38], are predicted to result in different ignition responses.

For example, it has been suggested that tropical peat, having higher wood content, is associated with higher calorific values and greater flammability than high-latitude peat[38]. Despite these observations, there has been no attempt to study the variation in resistance to ignition of peat at the global scale, nor contrast in detail differences between tropical and boreal peats. To fill this knowledge gap, we obtained 152 peat samples from 55 sites covering arctic, boreal, temperate and tropical regions, including surface peat from all regions, and subsurface samples from the extreme latitude groups (arctic and tropical regions) (Table S1; Fig. 1) and assessed both their botanical composition and flammability.

A pyrolysis combustion flow calorimeter is used to heat each sample at the same rate and measure the temperature at which the maximum rate of thermal decomposition ($T_{max}$) occurs in each type of peat. $T_{max}$ approximates the ignition temperature and is thus a key parameter in determining the potential of a surface flaming fire to cause ignition of the peat below. $T_{max}$ therefore serves as a measure of a material's thermal recalcitrance, because material that is more resistant to heat will require a greater heat flux and requires a greater temperature to be reached before it will ignite. We couple these data to a composition analysis, to determine the plant constituents forming the peat. We show that boreal and temperate *Sphagnum*-dominated peat is the least resistant to thermal decomposition whilst tropical peat is considerably more resistant to ignition. Moreover, removal of surface peat in tropical regions will lead to a reduction in material ignitability as deeper peat layers are more thermally recalcitrant. This increase in $T_{max}$ with depth in tropical peat should result in a negative feedback on increased smouldering fire activity.

## Results and discussion

### How variable is peat resistance to thermal decomposition?

Tropical peat displayed higher $T_{max}$ (mean 420 °C) than arctic (354 °C), boreal (345 °C) or temperate (351 °C) peat (Fig. 2). Differences between tropical and all other latitude groups were significant ($p < 0.001$), and differences between all extratropical latitude groups were not significant ($p > 0.6$) [1-way ANOVA; Tukey's pairwise]. In the tropical peat, mean $T_{max}$ was 403 °C for the upper, aerobic layer, compared to 431 °C for the lower, waterlogged, compacted layer, and this difference was significant ($p = 0.004$). Extratropical peat showed a mean $T_{max}$ of 343 °C for the aerobic layer and 354 °C for the anaerobic layer, and the difference was not significant ($p = 0.058$). This indicates that tropical peat requires a higher temperature (or greater heat flux) to reach peak pyrolysis rate and therefore ignition, than the temperate, boreal or arctic peats tested. The tropical peats tested therefore have intrinsically greater fire resistance than the peats we tested from higher latitudes, and the resistance of the tropical peats to thermal decomposition increases with depth. This change with depth was not evident in higher latitude peat.

### Drivers of resistance to peat ignition

The botanical composition analysis (Fig. 3) shows that tropical peat ($n = 20$) consisted primarily of wood, roots, and unidentifiable organic matter in varying proportions, with minor components (3–18%) of undifferentiated aboveground biomass. This is in keeping with the normal forest-based origin of peat in lowland tropical climates[40]. Sedges (Cyperaceae) were also present in samples from a single site, Oropel Swamp, Panama (48% in aerobic layer, 43% in anaerobic layer), which has the highest absolute latitude of the tropical group, and despite its tropical climate[41] may represent a transition to a subtropical peat composition, which is typically sedge- rather than tree-dominated[40]. Mosses were not present in any of the tropical samples, and no visually discernible differences in the botanical constituents were evident between aerobic and anaerobic samples. Temperate aerobic layer peat ($n = 12$) had *Sphagnum* as its primary component, except samples from Slieveanorra, Ireland (sedge peat)

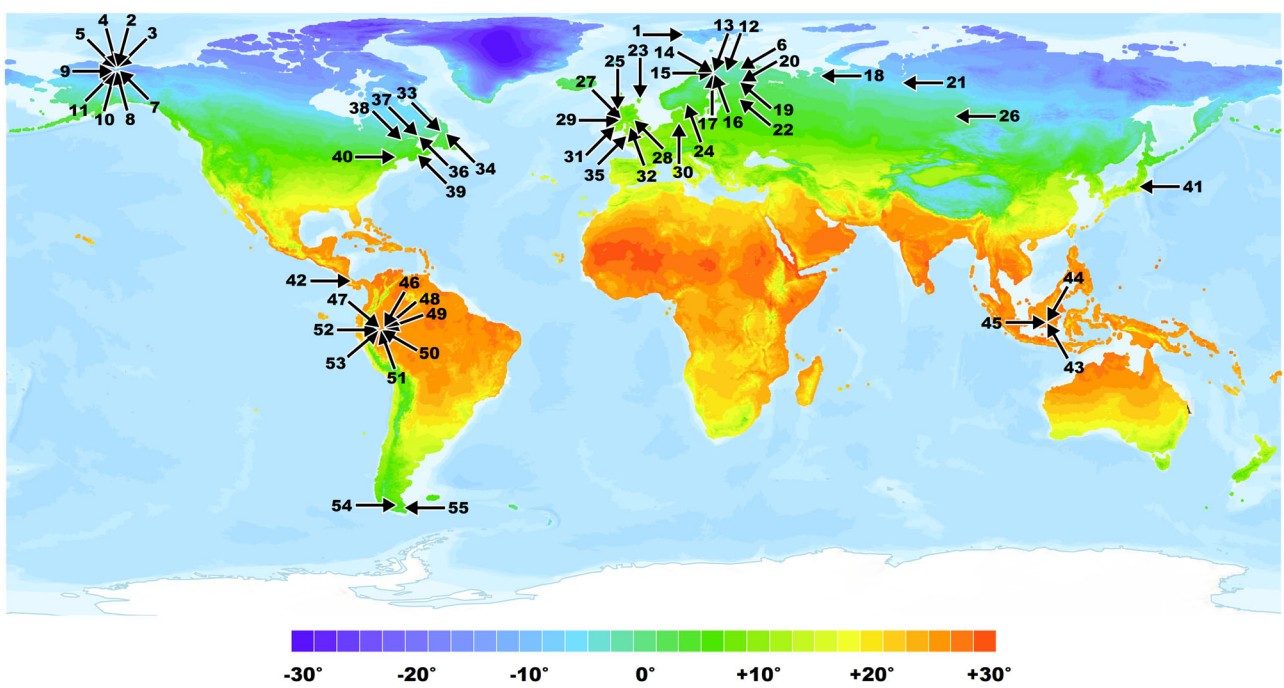

**Fig. 1 | Locations of the 55 study sites, showing mean land surface temperature 1961–1990 (ref. 82).** Site details are given in Table S1. Map created using ArcGIS® software by Esri. ArcGIS® and ArcMap™ are the intellectual property of Esri and are used herein under license. Copyright © Esri. All rights reserved. Basemap credits: Esri, USGS.

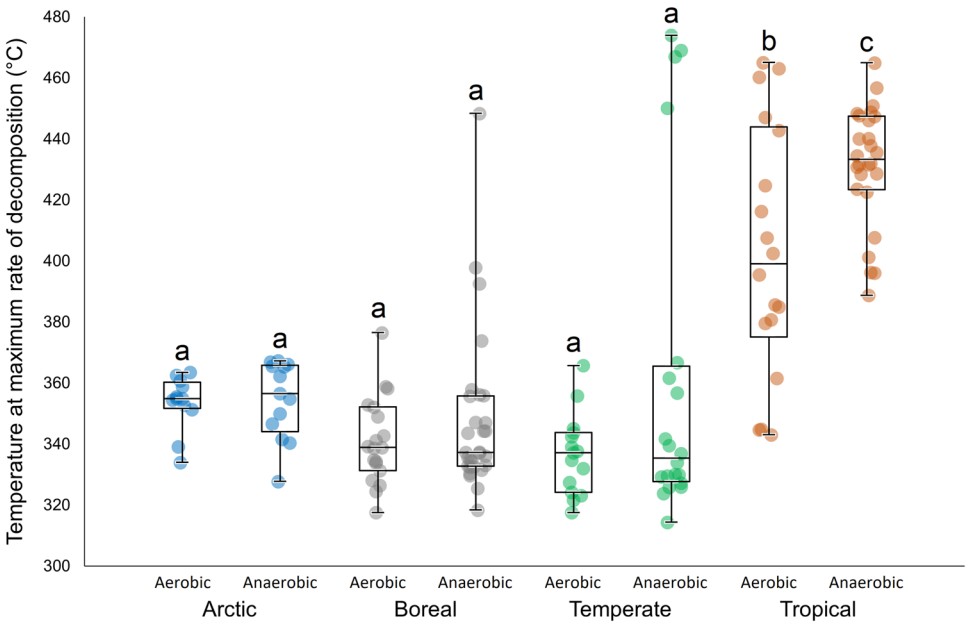

**Fig. 2 | Distributions of mean $T_{max}$ values for aerobic layer and anaerobic layer peat samples from different climatic regions.** Box plots display minima, 1st, 2nd and 3rd quartiles, and maxima. Different letters above the box plots indicate statistically significant differences at $p < 0.001$ (1-way ANOVA, Tukey's pairwise, $n = 152$).

and Shestakovo, Russia (wood and root peat) which contained no identifiable *Sphagnum* or other mosses. Sphagnum mosses are normally the dominant peat-forming vegetation in temperate regions[42]. The peat from Slieveanorra has previously been characterised as a *Sphagnum*-containing sedge peat[43], and it may be that its high degree of humification[43] had obscured some *Sphagnum* content in our samples. Shestakovo, in southern Siberia, experiences a highly continental climate[44] which is associated with the formation of moss-free peat substantially formed from the underground parts of sedges and other Poales[40]. The *Sphagnum*-containing peat of the other temperate

samples ranged from 41–94% mosses (26–94% *Sphagnum*) with varying additional components of which only roots were present in all samples. Boreal aerobic layer peat ($n = 12$), which also typically has *Sphagnum* as its main component[42], was more uniformly moss-dominated, with 49–98% mosses (43–98% *Sphagnum*). Arctic peat ($n = 9$) was variously dominated by *Sphagnum*, sedges, wood or roots, with some aerobic layer peats having high wood and root contents but their corresponding anaerobic layer samples having a higher sedge content, which may reflect a change in vegetation cover over time[45]. We found that while peat composition is heterogeneous within each latitude

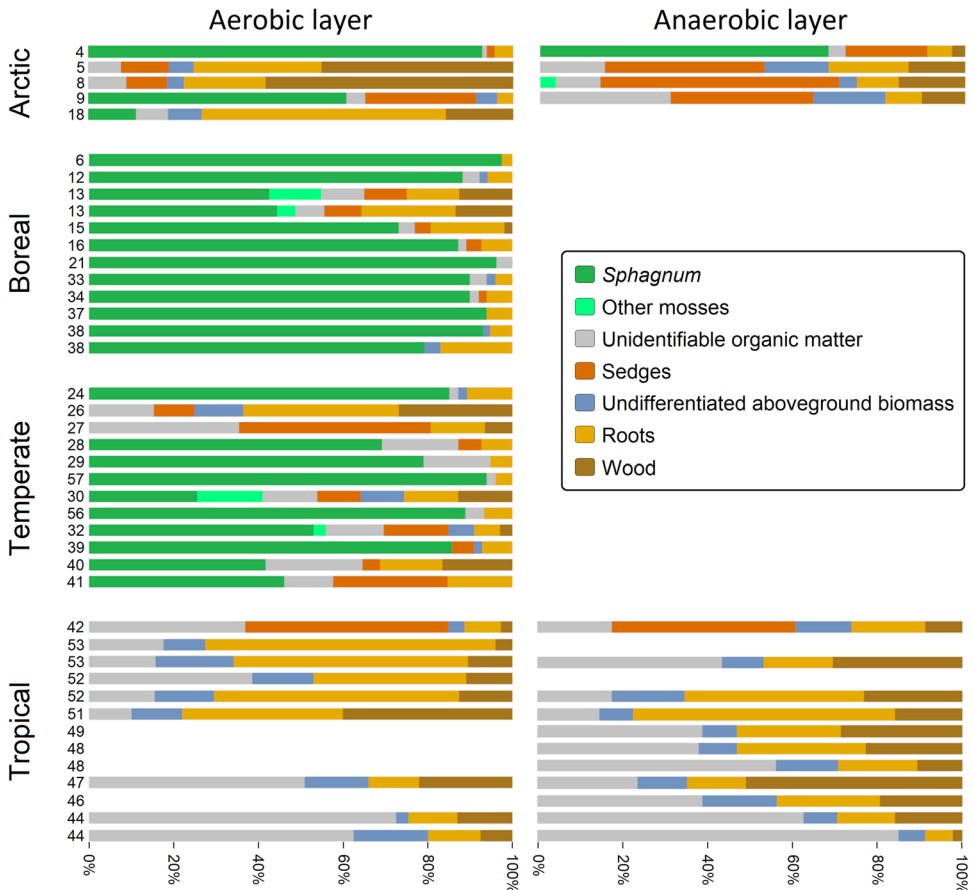

**Fig. 3 | Composition of aerobic layer and anaerobic layer peat samples from different climatic regions.** Sites are arranged from north to south within each region, and site numbers correspond to Fig. 1 and Table S1.

group, there is a clear latitudinal division between generally root/wood-dominated tropical peatlands, and generally *Sphagnum*-dominated extratropical peatlands, although this may reflect some sampling bias (for example, lack of *Papyrus*-dominated tropical peatland samples in our collection).

When grouped by peat composition, $T_{max}$ was markedly higher in humified (decomposed) peat samples (mean 414 °C) and wood/root peat samples (400 °C) than in sedge (360 °C), *Sphagnum*/sedge (341 °C), or *Sphagnum* (340 °C) peat samples (Fig. 4). $T_{max}$ differed significantly ($p < 0.005$) between humified and all other peat categories except wood/root ($p = 0.779$); wood/root also differed significantly from *Sphagnum* and *Sphagnum*/sedge peat samples ($p < 0.001$) but not from sedge peat samples ($p = 0.062$). This indicates that humified and wood/root peat requires greater heat fluxes to ignite than sedge- or moss-dominated peat. Across all five compositional categories, the samples with $T_{max} > 370$ °C originate exclusively from tropical climates, and those with $T_{max} < 370$ °C are mostly from higher latitudes, with only 2 of 35 samples being of tropical origin.

Relationships between $T_{max}$ and individual peat components are shown in Fig. 5. Correlations ($n = 53$) were quantified using Spearman's ρ due to the non-linear nature of the relations. $T_{max}$ was positively correlated with content of wood (ρ = 0.60, $p \ll 0.001$), roots (ρ = 0.44, $p = 0.001$), and undifferentiated aboveground plant remains (ρ = 0.68, $p \ll 0.001$), and negatively correlated with mosses (ρ = −0.66, $p \ll 0.001$). As shown in Fig. 5, both higher $T_{max}$ and higher wood, root and undifferentiated aboveground biomass content, are associated with tropical origin of the peat, which is also associated with zero moss content in all cases. This suggests a latitude or climate-based control on both composition and $T_{max}$. Relationships of $T_{max}$ to latitude and

climate are shown in Fig. 6 ($n = 152$). $T_{max}$ was negatively correlated with absolute latitude (ρ = −0.38, $p \ll 0.001$), and positively correlated with mean temperature (ρ = 0.41, $p \ll 0.001$).

Thermal recalcitrance of the tropical peat samples relative to the temperate, boreal or arctic samples likely results from differences in lignin/holocellulose ratios. Lignin content of plant matter is a determinant of chemical and biochemical recalcitrance[46,47] and thermal stability[48,49]. Tropical peat tends to be dominated by woody material[38,50], and therefore will have a high lignin content[51,52]. High-latitude peat samples are typically dominated by *Sphagnum* and Cyperaceae[50] and should therefore have a higher holocellulose content. As mosses do not contain lignified cell walls[53], peat composed almost entirely of *Sphagnum* will contain negligible lignin. Our composition analysis shows a very strong tendency toward tropical peat samples being wood/root-dominated, and temperate and especially boreal peat samples being *Sphagnum*-dominated. However, the arctic peat samples are of varying composition, although the majority are dominated by lignin-containing components.

That lignocellulose composition is important in determining $T_{max}$ is supported by the correlations of $T_{max}$ with different peat components. Across all samples, $T_{max}$ has a moderate positive correlation with wood content (ρ = 0.60, $p \ll 0.001$) (Fig. 5a) and root content (ρ = 0.44, $p = 0.001$) (Fig. 5b). Mosses, which are devoid of lignin, show a negative correlation with $T_{max}$ (ρ = −0.66, $p \ll 0.001$) (Fig. 5d), evidencing a binary relationship in which $T_{max}$ exceeds 370 only in the absence of moss. The difference in $T_{max}$ between tropical and higher latitude peat samples is likely a direct result of the high content of woody material, and thus lignin, in tropical peat samples. This is also supported by the slightly higher $T_{max}$ of the woodier arctic peat

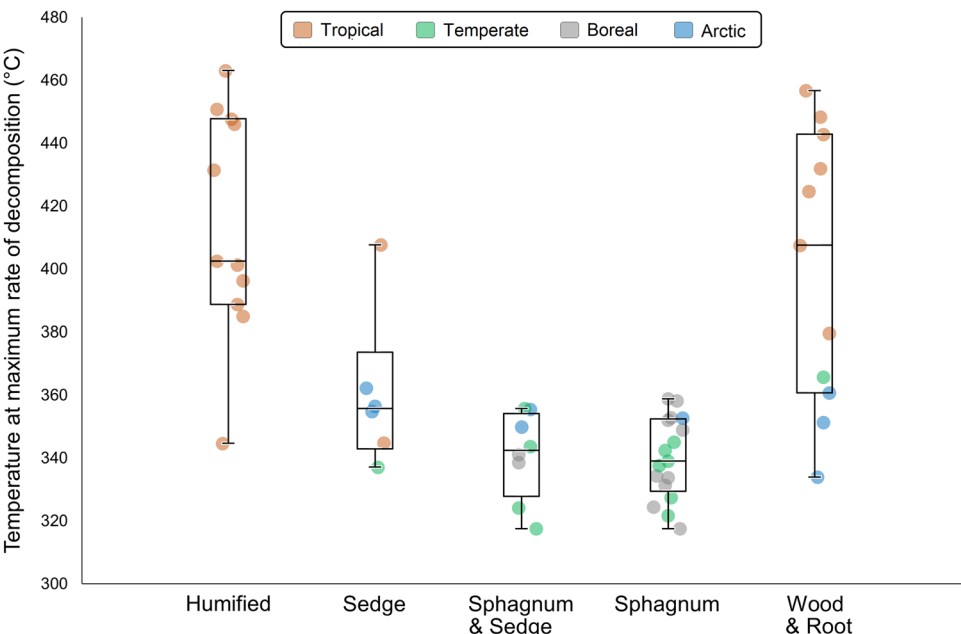

**Fig. 4 | Distributions of mean $T_{max}$ values for different peat composition types.** Box plots display minima, 1st, 2nd and 3rd quartiles, and maxima.

samples than the boreal or temperate ones. However, peat samples with high wood/root content nonetheless have lower $T_{max}$ where they originate from high-latitude sites (Fig. 4), and tropical peat samples generally have the highest $T_{max}$ regardless of peat type (Fig. 4). The distribution of our tropical and higher latitude peat samples across the five composition categories suggests that the climatic origin of the samples may be the controlling factor and that the relationship between $T_{max}$ and peat type arises from the fact that moss-dominated peatlands do not form in low altitude tropical conditions.

The presence of peatlands in tropical regions, despite higher temperatures leading to faster decomposition, may be due in part to their high woody content inhibiting decomposition[50]. Greater aromatic content (lignins, tannins, and humic substances) and lower carbohydrate content make tropical peat more chemically recalcitrant than arctic and boreal peat[50]. This is partly attributable to the high lignin content of woody parent vegetation, but also to higher temperatures enhancing humification so that an initial phase of decomposition leaves the peat in a more recalcitrant state[50]. Therefore, the high woody content of tropical peat may be due not only to predominantly woody vegetation but to selective incorporation of more ligneous material into the peat, whose chemical recalcitrance counteracts the effect of high temperatures on decomposition rates. While both tropical and some arctic peat may have high woody content, in a tropical environment the more labile components will have been preferentially decomposed, increasing the recalcitrance of the remaining material prior to preservation under anoxic conditions (while ligneous material maintains the structural integrity that allows the material to be identified as wood). Therefore, the high $T_{max}$ of tropical peat may largely represent a concentration of (bio)chemically recalcitrant material during decomposition, resulting in peat which also has higher thermal resistance (i.e. reduced ignitability). This interpretation is supported by the fact that $T_{max}$ is consistently lower (mean 341 °C) where moss is present in any quantity. As moss is the only peat component free of lignin, and peat samples vary widely in moss content, a direct effect of the lignin/holocellulose ratio on $T_{max}$ would be expected to show a more linear relationship as is seen for wood and root content. Instead, $T_{max}$ varies from ~320–370 °C if moss is present (which corresponds entirely to higher latitude sites), and approximately 330–470 °C where moss is absent (27 out of 33 sites

being tropical). This suggests that the relation between $T_{max}$ and moss content is likely to be a secondary effect of an underlying relationship between $T_{max}$ and tropical or extratropical peat origin.

## Implications for tropical forest peatland vulnerability to fire

Our results have important implications for understanding the vulnerability of drained tropical peats to ignition. Tropical peatlands are concentrated in southeast Asia (with 250,000 km² out of 400,000 km² [15,54]), where they have been subject to extensive drainage and deforestation in recent decades[54,55], and are especially at risk of fire[11]. In tropical peat swamp forests, peat preservation is dependent on forest cover, which stabilises the peat and maintains a moist microclimate, while the forest cover is itself dependent on the integrity of the peat, including for structural support and hydrological regulation[56]. Therefore primary peat swamp forest does not easily recover from disturbance, and secondary growth is susceptible to domination by ferns and shrubs with higher fire risk[26]. Fire may be employed for land clearance, as has occurred widely in the peat swamp forests of insular southeast Asia[6], which this century have seen drastic increases in fire frequency and severity[6], and lost coverage at a rate of 2.25% yr⁻¹ from 2000 to 2010[6]. Drainage and deforestation also reduce the fire resistance of the surrounding forest due to drawdown of the water table, commonly extending for several hundreds of metres from the forest edge[57,58], and effects on microclimate[6]. Increases in fire frequencies have been exacerbated by climatic changes[55].

In the peat swamp forests of southeast Asia, there appears to be both positive and negative feedback associated with fire. A greater frequency or severity of burning progressively reduces tree regrowth and shifts species composition toward a more flammable fern- and sedge-dominated community[6,59]. Therefore, an initial fire also results in greater subsequent ignition risk due to reduced humidity after the loss of tree cover[11], and changing fire behaviour due to fuel loading from dead but unburned trunks, and fallen trees resulting from loss of soil integrity[11]. Negative feedbacks also operate, in which fire frequency or severity may be reduced by the effects of previous fires. Several such feedbacks have been suggested in relation to fuel loading. Depletion of fine surface fuels can reduce surface fire intensity and may limit fire spread[60]. In peat swamp forests, where almost all aboveground biomass can be lost after repeated fires[11], a shift from tree cover to

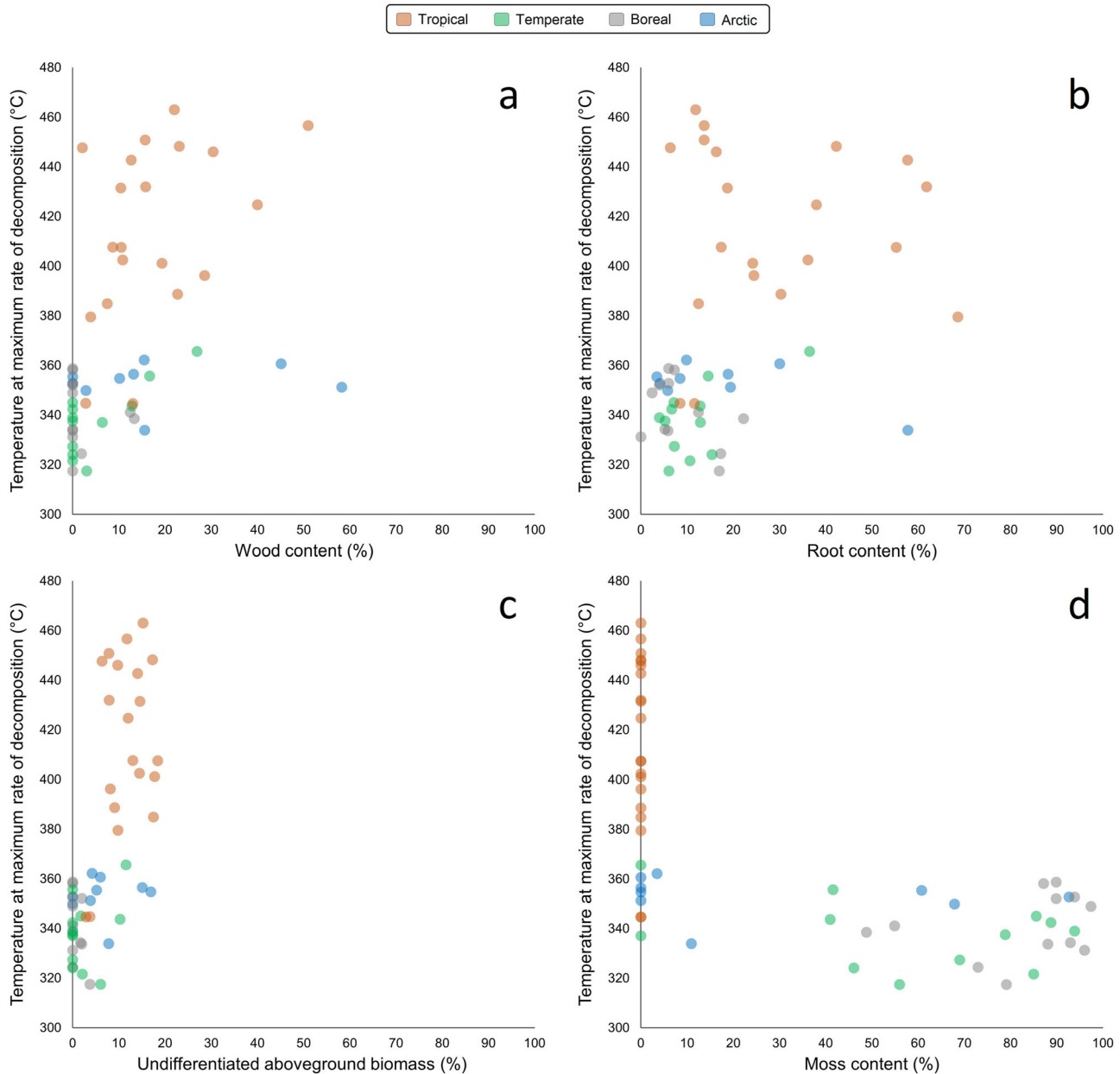

**Fig. 5 | Correlations between $T_{max}$ and content of different peat components. (a)** wood content, **(b)** root content, **(c)** undifferentiated aboveground biomass, **(d)** moss content.

non-woody vegetation is likely to prevent surface fires transitioning to peat fires, which require greater fuel load and temperature[11]. However, we note that where fires are started deliberately, with the aim to burn vegetation, accelerants will often be used, which may alter the heat flux delivered to fuel components including those of the peat.

It has been noted that in the area of the former Mega Rice Project[61] in Kalimantan, Indonesia, degraded peat swamp forest appears to show decreasing depth of burn with subsequent fires, and a range of possible reasons have been suggested[6]. Firstly, this may be due to progressive reduction of the aboveground fuel load. Secondly, the loss of peat, which can typically be to depths of 30 cm or more[62,63], reduces the distance to the water table, thus effectively increasing peat moisture content. Thirdly, the post-fire peat surface is left more recalcitrant as a consequence of selective destruction of more labile forms of carbon such as lignins and polysaccharides, and accumulation of aromatic and aliphatic compounds[64]. The increase in $T_{max}$ with depth found in the present study suggests an additional

negative feedback linked to the botanical composition of tropical peat. Our results for the global variation in $T_{max}$ indicate that tropical peat requires heating to a higher temperature via a greater flux of heat from a surface fire to ignite. Our results also indicate that fire resistance further increases with depth in tropical peats (Fig. 2). This suggests that if increases in fire severity, or shorter fire return intervals, were to cause the surface layers of peat to be lost at a greater rate than that of peat accumulation, then exposure of more ignition-resistant peat ought to decrease its subsequent vulnerability to fire. Whether this negative feedback influences the prevalence of peat fires will depend on the temperatures attained at ground level due to the heat flux from the burning of overstory vegetation. If these far exceed the ignition temperature of the peat, variation in that temperature should not be relevant to the probability of ignition. Yet where the heat flux delivered means that ground temperatures fall within the range of peat ignition temperatures, variation in the latter may determine ignition.

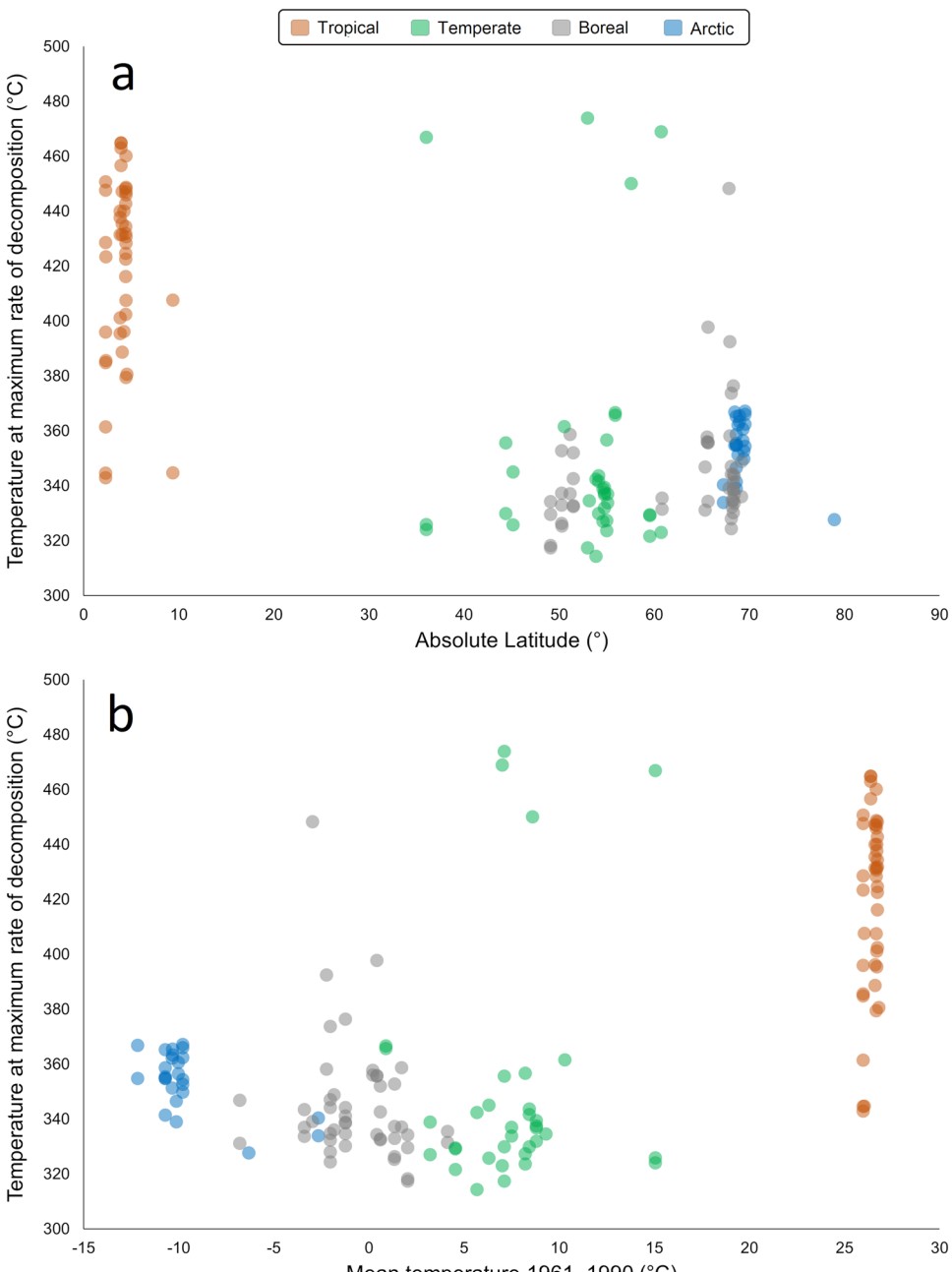

**Fig. 6 | Correlations of mean $T_{max}$ values with absolute latitude and mean surface temperature 1961-1990 (ref. [82]). a**: absolute latitude, **b**: mean surface temperature 1961–1990 (ref. [82]).

Ground temperatures attained due to surface fire will vary spatially and temporally, and ignition of the peat will depend on the temperature profile (i.e. temperature as a function of time) as the fire passes, with the required heat input being substantially affected by moisture content[34]. Since surface fire may or may not ignite the underlying peat[21,39], ignition temperature is likely to be a determining factor, with an inverse relationship to ignition probability. This suggests that the loss of upper peat layers (with lower ignition temperatures) does have the potential to leave a peat surface with a lower likelihood of supporting the transition from surface fires to ground fires.

However, the presence of twigs and roots in peat has been reported to promote the propagation of smouldering fires[65,66], and it has been suggested that larger wood pieces in tropical peatlands can serve as conduits for conducting surface fires into the ground,

assisting flaming surface fires to transition to smouldering ground fires[37]. Moreover, woody pieces can result in gaps within the peat[37] that allow oxygen ingress, enhancing smouldering. Therefore it is likely that wood particles of different size affect flammability in different ways. Small woody particles would tend to lead to overall high lignin contents of peat at the scale measured in this study, reducing ignitability, whilst larger particles may assist with carrying fire and oxygen down into the ground. Hence the vulnerability to smouldering ignition of tropical peat will be decreased by higher wood content, requiring a suitably intense ignition source that allows exposed peat to reach a temperature sufficient to drive pyrolysis and initiate the process of char oxidation (smouldering); but once ignited, larger wood particles may enhance the ability of the fire to propagate through the peat[67].

Higher calorific values (19–23 kJ g$^{-1}$) have been recorded for peat from Indonesia when compared with boreal peat[38]. This should

increase the energy release per equal peat volume, drying the peat and enhancing pyrolysis at the smouldering front. Moreover, non-*Sphagnum* based peat tends to have a higher bulk density[68,69], which should produce more energy during smouldering due to the oxidation of a greater mass of peat particles[70]. However, moisture content strongly interacts with bulk density (where higher bulk density peat holds more water) to determine whether or not smouldering propagation can be maintained[34,67]. High-density peat, with generally higher moisture content, usually either fails to sustain smouldering or tends to carry slower fires[34]. However, the moisture content of degraded peat in Sumatra has been observed to be as low as ~20% dry mass in the uppermost 10 cm of the peat[71], and where there is little water to act as a heat sink, high-density peat will instead provide a large energy source that will support self-sustained propagation of smouldering fires. However, measurements of peat from the former Mega Rice Project area of Kalimantan[61] have shown that calorific content appears to decrease with depth[72]. This coupled to the increase in $T_{max}$ in the anaerobic layers of tropical peat ought to lower the risk of ignition and decrease the potential of self-sustained propagation with depth in cases where previous fires have removed the surface peat.

In summary, we conclude that where peat has been drained and degraded, consideration of the botanical composition of peat may have more importance in determining its flammability than has previously been considered. Our findings, which link the botanical constituents of peat and its resistance to ignition, indicate that the composition of tropical peat confers resistance to ignition, but also propensity for more intense burning when in a dry and degraded state. The higher wood content, calorific content and bulk density in tropical peat when compared with peat of other latitude groups will tend to decrease its ignitability but increase the energy release per equal peat volume once smouldering is initiated, allowing self-sustained propagation through the peat. However, we suggest that deeper tropical peat tends to have a lower vulnerability to fire since both the increase in $T_{max}$ and decrease in calorific content will have negative feedback on subsequent ignition and self-sustained propagation of smouldering within the peat.

This study of peat types from around the world provides a first attempt at assessing how latitudinal effects and their influence on botanical composition may influence flammability. Further research might seek to determine in more detail how variable ignitability and calorific content are across specific degraded peatland systems. If estimates can be made as to their intrinsic flammability this may serve as a predictor for future fire severity and/or aid in determining which areas might need more focused fire protection and ecosystem restoration than others.

## Methods

Peat samples were obtained from 55 different sites: 11 arctic, 16 boreal, 16 temperate and 12 in tropical regions (Table S1; Fig. 1). At each site, samples were taken from between one and four coring locations, depending on existing research designs. Sample depths were determined by total peat depth at each location, with surface (aerobic layer) samples collected from depths of 0–15 cm, and subsurface (anaerobic layer) samples from 19–468 cm. 152 samples were obtained for calorimetric analysis, and 53 of these were additionally analysed for composition.

### Calorimetric analysis

All samples were analysed by pyrolysis combustion flow calorimetry (PCFC)[73], in which the oven-dried sample is pyrolysed in a nitrogen stream, the resulting pyrolysate gases fully combusted in excess oxygen, and the heat release measured by oxygen consumption calorimetry. The sample is thermally decomposed at a constant rate of temperature rise, and the heat release profile from combustion of the pyrolysate used to derive flammability metrics. Whereas flammability parameters obtained from conventional combustion experiments are affected by initial differences (or in-process changes) in the size and shape of the specimens, and by edge effects, PCFC measures intrinsic material properties that are independent of test conditions[74], resulting in a high degree of reproducibility. $T_{max}$ is the temperature at which the maximum rate of solid mass loss, and thus of pyrolysate generation and heat release, is attained, and is approximately equal to the ignition temperature of the material[75]. We used an FAA Micro Calorimeter (Fire Testing Technology Ltd, East Grinstead, UK), which is designed for testing the flammability of construction and furnishing materials, but has recently been used to assess combustion properties of wildland fuels[76–79]. Each peat sample was analysed in duplicate (sometimes triplicate), using subsamples of 1.2–29.9 mg. The pyrolyser heating rate was $3\,°C\,s^{-1}$, the maximum pyrolysis temperature 750 °C, and the combustor temperature 900 °C. The $N_2$ flow rate was 80 cm³ min⁻¹ and the $O_2$ flow rate 20 cm³ min⁻¹. The experimental method follows ASTM D7309-07 Method A[80]. $T_{max}$ values for paired replicates, representing separate calorimetric tests of material from the same peat sample, were highly correlated ($r = 0.97$), indicating that the test produces replicable results.

### Compositional analysis

Peat samples for composition analysis were sieved through a 125 μm sieve using a spray of deionised water. The material retained on the sieve was then analysed for peat components and macrofossils following a standard protocol[81]. Samples were placed in a petri dish and scanned using a low power (×10–×50) stereo-zoom microscope with a 10 × 10 square grid graticule inserted into one of the eyepieces. The petri dish was moved randomly to 15 different views, plant macrofossil types were estimated as percentages for each view using the graticule, and the results were averaged to represent the whole sample. High power (×200–×400) microscopy was used to confirm identifications. Subsamples of plant macrofossil material were mounted on microscope slides (temporary preparations using water as a mountant) and identified at ×100–×400 magnification. The samples were categorised into broad composition types (*Sphagnum* peat, sedge peat, *Sphagnum* and sedge peat, wood and root peat, and humified peat). To avoid bias, the composition analysis was undertaken without knowledge of the sample origins.

Climate (temperature) data[82] were obtained via the AQUASTAT Climate Information Tool[83]. Temperatures obtained for each site are mean values for the period 1961–1990, interpolated from climate station data at a spatial resolution of 10 min[82].

## Data availability

The data supporting the findings of this study are available at: https://doi.org/10.6084/m9.figshare.25858402.

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

## Acknowledgements

We thank Matthew J. Amesbury, Michelle Garneau, Mark J. Grosvenor, Victoria A. Hudspith, Markku Mäkilä, Lisa Orme and Nicole K. Sanderson for providing peat samples. C.M.B., S.P. and A.G-S. acknowledge funding from the KaLi Project, funded by the UKRI Global Challenges Research Fund, Grant No. NE/T010401/1. A.G-S. has also received funding from NERC (NE/I012915 and NE/S001166/1) and from the European Research Council (ERC) under the European Union's Horizon 2020 research and innovation programme (grant agreement No 865403). This work reflects only the author's view and the European Commission/Agency is not responsible for any use that may be made of the information it contains. G.T.S. has received funding from the Dutch Foundation for the Conservation of Irish Bogs, The Quaternary Research Association and Leverhulme Trust RPG-2021-354. T.A.B. acknowledges funding from Russian Science Foundation Grant N 23-27-00217.

## Author contributions

C.M.B. and A.G-S. designed the study. A.G-S., G.T.S., T.A.B., H.C-Q., D.J.C., M.G., P.D.M.H., O.L., D.M., S.P., T.P.R. and M.V. obtained peat samples. C.M.B. and S.N. conducted calorimetry experiments. G.T.S. analysed peat composition. A.J.C. analysed data with input from C.M.B. and A.G-S. A.J.C. and C.M.B. wrote the text with input from all authors.

## Competing interests

The authors declare no competing interests.
