## [Peer Review File · Nature Communications]

Tropical peat composition may provide a negative feedback on
fire occurrence and severityReviewer #1 (Remarks to the Author):

this study looks at the temperature of maximum thermal decomposition in peatlands from around the world and found that tropical peatlands (both surface and deep peats) are distinct from the other extra-tropical peatlands as their T_{max} is much greater, suggesting that they might burn less easily (if fires were to occur). I enjoyed reading this manuscript and liked the descriptive style of the author. I have a few minor comments below, which once addressed, may improve the way some of the results are presented.

This manuscript left me wondering how 'hot' fires burn in tropical vs. extra-tropical regions (is there a difference)? what is the range of ground burning temperatures across the globe? It could be important to mention this information.

Figure 5: Botanical composition / differences across latitudes: those r² values are quite high, but looking at the plots leaves me a bit perplex. The correlations really just exist because the tropical peats are different from the extra-tropical peatlands. Taken individually, the % of these peat components really doesn't correlate much with T_{max}. can you add a comment to this effect in your discussion?

Figure 6; similar comment as above. The regression lines are misleading, as what the plots show is that there are 2 distinct groups of peats: tropics vs. extra-tropics.

I appreciate the authors' discussion on the many (and opposite) feedback loops and processes impacting the severity of peat fires. But I was wondering if it would be possible (space allowing) to add some sub-sections to your Discussion, specifically to separate out 'your' results and findings about peat quality vs. the 'other' findings, to make it easier for the reader to follow your argument. I also think that slipping up the text into sub-sections would strengthen your overall argument, which gets lost a little bit in all the details.

Reviewer #3 (Remarks to the Author):

Nature Comms Review 23-40783

General:

This paper analyses the botanical composition and thermal decomposition properties of global peat samples. The findings are discussed mainly in the context of controls on tropical peat fires. Overall the content is interesting and relevant to the wider field as globally the peatland carbon sink is under threat. I find the article in its current state unbalanced, I would suggest reframing the paper to either focus on tropical peat flammability – testing tropical against the rest of the world, or balance the writing to discuss all biomes. The authors appear to introduce three potential hypotheses in the “implications” section. Consider using this to frame the paper, either for tropical peats as is currently written, or comparing the strength of each of the controls/feedbacks identified in each biome based on the botanical composition and thermal decomposition properties assessed.

Introduction:

While I can see the structure of the introduction narrowing down from global scale to small-scale physics and peat biogeochemistry I find the paragraphs disjointed. In particular, the continued jumping between boreal and tropics references confusing. I think some restructuring would help guide the reader to the specific question being addressed. Perhaps focussing on processes and controls rather than statements.

E.g., Line 57, add detail about how warming actually alters moisture (i.e., relative humidity), and therefore flammability, and whether to what degree this is a relevant process in the tropics, temperate and arctic regions.

Line edits:

Abstract: Line 18: “Should lead to a reduction in ignitability” – after the reading the paper and discussion of the other 3+ feedbacks influencing flammability and smouldering propagation I think

this needs to be reworded to reflect the potential (previously under-considered) influence of botanical/chemical properties in addition to the other controls.

Line 47: Also see Wilkinson et al recent paper on peat fire and drainage.

Line 51: wording – “exposes flammable peat” – as mentioned, it’s only flammable below a certain moisture content. Consider rewording to reflect this.

Line 57: The Siberia sentence seems disconnected. What about the warming has caused the increased fire activity? There are missing processes here.

Line 59+: Some numbers i.e., km², would be useful here.

Line 91: This paragraph is referring only to tropical peat ignitions being mainly anthropogenic in nature, I presume, but the narrowing down geographically is not clear to readers who don’t have in-depth knowledge of peatland fire.

Line 110: What constitutes surface and subsurface samples?

Line 115+: Should these results be held back for the results section?

Line 116: Are thermal decomposition and ignition used interchangeably throughout? Please clarify.

Results and Discussion:

Line 125: Figure 2 – referenced incorrectly in text. Swap 2 and 3.

Figure 3: add significance letters

Line 155+: $p << 0.001$ would suffice

Figure 5: negative r ? Also is this r^2 ?

Figure 6: include equations of line if relationship is significant

Can you comment on the variety of peatlands across the boreal (for example) and how that might influence your results, as you mention you have sampled from mainly Sphagnum dominated peatlands. Does the sedge dominated site differ from other sedge dominated samples? I presume this is difficult to answer because of the unknown botanical composition of 2/3 of the samples?

It would be helpful to read some measures of peat loss from tropical peat fires, is this 5cm or 50cm this will strongly influence the difference in moisture content through proximity to water table, and therefore influence the relative importance of the feedback you identify.

Line 264: This is an interesting complex feedback you lay out but it should be written in a more speculative way since you didn’t test this.

Methods: What’s the breakdown of the 53 composition samples?

Can you find a more recent climate record given that a lot of your samples are surface samples and are likely to have been influenced by the climate in the last 30 years and the climate in most regions has changed significantly over the last 30 years.

Reviewer #1 (Remarks to the Author):

this study looks at the temperature of maximum thermal decomposition in peatlands from around the world and found that tropical peatlands (both surface and deep peats) are distinct from the other extra-tropical peatlands as their T_{max} is much greater, suggesting that they might burn less easily if fires were to occur). I enjoyed reading this manuscript and liked the descriptive style of the author. I have a few minor comments below, which once addressed, may improve the way some of the results are presented.

This manuscript left me wondering how 'hot' fires burn in tropical vs. extra-tropical regions (is there a difference)? what is the range of ground burning temperatures across the globe? It could be important to mention this information.

Reliable data on this are difficult to obtain. Neither remote sensing imagery nor post-fire assessment can give an accurate ground temperature, which can only be obtained by instrumentation of an area before a fire (which essentially limits data collection to mainly prescribed fires). We are not aware of any available data that could be used to contrast tropical and extratropical peat fires in this way.

Figure 5: Botanical composition / differences across latitudes: those r^2 values are quite high, but looking at the plots leaves me a bit perplex. The correlations really just exist because the tropical peats are different from the extra-tropical peatlands. Taken individually, the % of these peat components really doesn't correlate much with T_{max} . can you add a comment to this effect in your discussion?

We accept that the correlations exist largely because of the differences between tropical and extratropical peats, but we do not think this makes their inclusion here inappropriate.

The variation in T_{max} that is evident in our global dataset is most directly explained in terms of composition (since it is a material property), and the plots in Figure 5 (ignoring the colour coding) present the relevant relationships in a clear and conventional way. As we seek the explanation for the differences in composition by looking at the environments the peats formed in, this is also indicated in Figure 5, and this reveals that the latitude group largely explains the correlations between T_{max} and composition. This point may have been lost due to the fact that we prioritise the relation of T_{max} to latitude in our discussion. We have aimed to clarify this by revising the text (Lines 144-162): first presenting the relations of T_{max} to composition, then identifying the latitudinal influence on both, and then defining the relation of T_{max} to latitude.

We acknowledge that the regression lines shown were not appropriate here, as we did not intend to suggest that any of the relationships shown in Figures 5 and 6 are necessarily linear (and some clearly are not). These were included as we used the associated Pearson's r as our measure of correlation. Given the non-linearity in the relationships, we have reassessed the correlations using Spearman's rho as the measure, and removed the regression lines. (The revised correlation values do not affect any part of our argument.)

Figure 6; similar comment as above. The regression lines are misleading, as what the plots show is that there are 2 distinct groups of peats: tropics vs. extra-tropics.

As per the above reply, the regression lines have been removed.

I appreciate the authors' discussion on the many (and opposite) feedback loops and processes impacting the severity of peat fires. But I was wondering if it would be possible (space allowing) to add some sub-sections to your Discussion, specifically to separate out 'your' results and findings about peat quality vs. the 'other' findings, to make it easier for the reader to follow your argument. I also think that slipping up the text into sub-sections would strengthen your overall argument, which gets lost a little bit in all the details.

We have improved the narrative flow of the results and subdivided the results and discussion under three headings. The first two discuss the overall findings of our analyses under the headings - *How variable is peat resistance to thermal decomposition* and *Drivers of resistance to peat ignition*. We then include a section specific to discussion based around tropical peatlands owing to the finding that the composition of tropical peats may have important considerations for this ecosystem's vulnerability to fire. This is entitled *Implications for tropical forest peatland vulnerability to fire*. We believe now that this presents a clearer structure.

Reviewer #3 (Remarks to the Author):

Nature Comms Review 23-40783

General:

This paper analyses the botanical composition and thermal decomposition properties of global peat samples. The findings are discussed mainly in the context of controls on tropical peat fires. Overall the content is interesting and relevant to the wider field as globally the peatland carbon sink is under threat. I find the article in its current state unbalanced, I would suggest reframing the paper to either focus on tropical peat flammability – testing tropical against the rest of the world, or balance the writing to discuss all biomes.

We think that reframing the study in the way suggested would misrepresent the research process that was actually undertaken and therefore also our findings. The core aim of our study was to explore to what extent variations in peat composition might control flammability, where the normal peat moisture contents that protect it from fire are lacking. With this in mind we sought to test as many peats as possible from across the globe. We found that where peat has been drained and degraded, consideration of the botanical composition of peat may have more importance in determining the flammability of peat than has previously been considered. But of importance our study also revealed that the composition of tropical peats confers resistance to ignition, but also propensity for more intense burning when in a dry and degraded state. To address the reviewer's concerns we have improved the narrative throughout the manuscript. We have firstly made the introduction clearer and removed any text that distracted from the main objectives of the study. Secondly, we have added headings within our results and discussion that we believe better explores in an ordered fashion our findings and better balances the manuscript. Finally, we have edited the abstract such that it captures our core aims from the onset.

The authors appear to introduce three potential hypotheses in the “implications” section. Consider using this to frame the paper, either for tropical peats as is currently written, or comparing the strength of each of the controls/feedbacks identified in each biome based on the botanical composition and thermal decomposition properties assessed.

We disagree that these were presented as hypotheses, but we have edited the manuscript to better reflect the discussion around variables that might influence the flammability of the tropical peats alongside that which this study itself has found. The edits can be seen primarily in the section entitled 'implications for tropical forest peatland vulnerability to fire'. We believe that we have altered the language and structure of the results and discussion to reflect that and that the headings now added to the results and discussion improve the clarity of the aims, findings and the importance of the difference we found in tropical peats.

Introduction:

While I can see the structure of the introduction narrowing down from global scale to small-scale physics and peat biogeochemistry I find the paragraphs disjointed. In particular, the continued jumping between boreal and tropics references confusing. I think some restructuring would help guide the reader to the specific question being addressed. Perhaps focussing on processes and controls rather than statements. E.g., Line 57, add detail about how warming actually alters moisture (i.e., relative humidity), and therefore flammability, and whether to what degree this is a relevant process in the tropics, temperate and arctic regions.

We agree that our introduction did seem somewhat disjointed in parts. We have now revised and improved the narrative and removed superfluous text that did not serve to direct the reader as to the background, aims and objectives our study. We believe that the introduction now flows much better and thank the reviewer for noting that this could be improved.

Line edits:

Abstract: Line 18: "Should lead to a reduction in ignitability" – after the reading the paper and discussion of the other 3+ feedbacks influencing flammability and smouldering propagation I think this needs to be reworded to reflect the potential (previously under-considered) influence of botanical/chemical properties in addition to the other controls.

We have amended this to "may lead to..." (Line 20). We also refer to the proposed feedback as being a "*potential negative feedback*" (Lines 21-22) for the same reason.

Line 47: Also see Wilkinson et al recent paper on peat fire and drainage.

This paper is now cited (Line 47).

Line 51: wording – "exposes flammable peat" – as mentioned, it's only flammable below a certain moisture content. Consider rewording to reflect this.

In the context of the previous lines, 'flammable' here means not prevented from ignition by its moisture content. We have amended the text to clarify this (Lines 50-51).

Line 57: The Siberia sentence seems disconnected. What about the warming has caused the increased fire activity? There are missing processes here.

The suggestion here was that the increased fire activity resulted from changes in the water table, and so this was a further (though non-essential) example of the processes described in lines 49–55. However we note that the papers cited do not establish this definitively, and so we have removed these references.

Line 59+: Some numbers i.e., km², would be useful here.

We have added data on land coverage here. (This paragraph has now been moved to the Discussion, and the data is at Lines 213-214.)

Line 91: This paragraph is referring only to tropical peat ignitions being mainly anthropogenic in nature, I presume, but the narrowing down geographically is not clear to readers who don't have in-depth knowledge of peatland fire.

We have amended this to "Peat combustion is normally initiated by the heat flux from a flaming vegetation fire, which in tropical regions is often anthropogenic" (Lines 70-71).

Line 110: What constitutes surface and subsurface samples?

This is explained in the first paragraph of the Methods, and in Table S1. We have amended the text at Lines 318–320 to make clear that 'surface' refers to the aerobic layer, and 'subsurface' to the anaerobic layer.

Line 115+: Should these results be held back for the results section?

We have followed the format commonly used in this journal. The last three sentences of this paragraph (Lines 94-98) could be removed, but we feel they are useful in signposting the key points of the Discussion section.

Line 116: Are thermal decomposition and ignition used interchangeably throughout? Please clarify.

We do not think we say anything that would suggest these terms are used interchangeably. The relationship of the two terms is explained in the last paragraph of the Introduction (and also mentioned in the Methods) and the usage elsewhere is consistent with this statement. T_{\max} is the calorimetric property investigated in this study, and its relevance is that it approximates the ignition temperature. Both terms are used according to their exact meanings.

Results and Discussion:

Line 125: Figure 2 – referenced incorrectly in text. Swap 2 and 3.

Figure numbers have been corrected.

Figure 3: add significance letters

These have been added.

Line 155+: $p \ll 0.001$ would suffice

These have been changed.

Figure 5: negative r ? Also is this r^2 ?

In response to comments from Reviewer #1, we now use the nonparametric ρ as the measure of correlation. It is negative for the relationship between moss content and T_{\max} , as discussed at Lines 156-157 and 177-179.

Figure 6: include equations of line if relationship is significant

In response to comments from Reviewer #1, we have removed the regression lines, which were included in reference to the associated r values (now replaced with ρ as per the above reply) and not intended to accurately model the relationships shown in the scatterplots.

Can you comment on the variety of peatlands across the boreal (for example) and how that might influence your results, as you mention you have sampled from mainly Sphagnum dominated peatlands.

We have rewritten parts of the text and added additional references, between Lines 116–143, to outline how our peat composition data compare to expectations based on general peat composition in the different regions. Briefly:

- Tropical peat is wood/root-dominated as expected (cf. response to next comment).
- Temperate peat is typically Sphagnum-dominated. We have added text to discuss the two temperate samples that were not.
- Boreal sites are all moss-dominated as is expected.
- The arctic peat samples could be biased toward an unrepresentative area, with an absence of moss in several samples being potentially atypical.

Note that this potential bias does not affect our overall arguments, since it would reduce the difference in composition between tropical and extratropical peats. We are not able to quantify how representative of the peat types within each latitude group our sites are; to our knowledge there is no comprehensive quantitative assessment of the prevalence of different peatland types with which to compare.

Does the sedge dominated site differ from other sedge dominated samples? I presume this is difficult to answer because of the unknown botanical composition of 2/3 of the samples?

We assume this refers to Site 42 (Oropel Swamp, Panama), which is the only sedge-containing peat from the tropical sites (all other latitude groups having several).

As can be seen in Figure 4, the aerobic-layer sample from Oropel has a T_{\max} within the range of values for extratropical sedge-containing peats, while the anaerobic-layer sample has a higher value close to the average for other tropical (sedge-free) peats. We would clearly need more data to establish whether either of these is representative. We do not think this is problematic, since the study is aimed at establishing global patterns by including many sites but with few samples from each: there are only two T_{\max} values for tropical sedge peat precisely because it is rare. Oropel Swamp is the highest-latitude of the tropical sites, and appears to display some characteristics of subtropical peat. It can be seen in Figure 6 that the two samples from this site are somewhat separated from the other tropical samples by latitude, but not by mean temperature – the site has a tropical climate. (There is no subtropical group in our data set, since subtropical peatlands are scarce globally, and unlikely to be represented in such a global data set unless specifically targeted.) We have added the following text at Lines 120–122 to briefly explain why this site appears as an outlier among the tropical sites:

[Oropel Swamp] “has the highest absolute latitude of the tropical group, and despite its tropical climate⁵³ may represent a transition to a subtropical peat composition, which is typically sedge- rather than tree-dominated⁵⁴.”

It would be helpful to read some measures of peat loss from tropical peat fires, is this 5cm or 50cm this will strongly influence the difference in moisture content through proximity to water table, and therefore influence the relative importance of the feedback you identify.

We have amended the text at Lines 243–244 to include this information, though there will be wide variation between different fire episodes:

“the loss of peat, which can typically be to depths of 30 cm or more, reduces the distance to the water table”

Line 264: This is an interesting complex feedback you lay out but it should be written in a more speculative way since you didn’t test this.

The statement of the feedback (at Lines 251–254) is presented as a suggestion (“This suggests...”), and this is followed by a paragraph discussing why it might not (always or ever) occur. We have made some further minor alterations to underscore the speculative nature: see also Lines 247-249 and Lines 304-307. We have also changed “should lead to a reduction in ignitability” to “may lead to a reduction...” in the abstract at Line 20.

Methods: What’s the breakdown of the 53 composition samples?

Details of the 53 samples are given in Figure 2, and details of the site locations in Table S1.

Can you find a more recent climate record given that a lot of your samples are surface samples and are likely to have been influenced by the climate in the last 30 years and the climate in most regions has changed significantly over the last 30 years.

We do not think it would be helpful to replace this climate data with another set, since while parts of our data set will derive from peats formed subsequently, others will have formed during the period of the climate data, and the anaerobic-layer samples long before. We make no attempt to relate specific peat samples to climatic conditions at the time of their formation (nor indeed to date them). Instead, we are only using the climate data in relation to much broader climatic differences than the variation of the last few decades. For this purpose, we think the FAO data is as valid as any alternative data set.

Reviewer #3 (Remarks to the Author):

Review 451299_1

Overall, the paper is much improved and an enjoyable read. The arguments are clear and logical, as are the figures. The framing of the paper is appropriate and the new findings presented are relevant to a wide audience.

Abstract

Line 22: "... reduced fire activity", this differs from your title "...reduced fire severity". Please align these two statements based on which you think most accurately describes your findings.

Intro:

The introduction is clear and has a good flow.

Results/discussion:

Line 135: should "is" be "was expected..." here?

Line 308-309: I find the "and degradation" part of the sentence is awkward. Consider re-phrasing.

REVIEWERS' COMMENTS

Reviewer #3 (Remarks to the Author):

Review 451299_1

Overall, the paper is much improved and an enjoyable read. The arguments are clear and logical, as are the figures. The framing of the paper is appropriate and the new findings presented are relevant to a wide audience.

Abstract

Line 22: "... reduced fire activity", this differs from your title "...reduced fire severity". Please align these two statements based on which you think most accurately describes your findings.

Response: We have changed both to "occurrence and severity". ('Activity' was intended to include both occurrence and severity but was admittedly unclear.)

Intro:

The introduction is clear and has a good flow.

Results/discussion:

Line 135: should "is" be "was expected..." here?

Response: The phrase "is also expected to have Sphagnum as its main component" was intended to convey the fact that these peats are usually Sphagnum-dominated, rather than being a prediction about the samples obtained for this study. We have changed this to "typically has Sphagnum as its main component" for clarity.

Line 308-309: I find the "and degradation" part of the sentence is awkward. Consider re-phrasing.

Response: This was intended to refer to the differential degradation of peat according to parent vegetation. This isn't necessary in this summary paragraph, so we have removed these two words rather than expanding the text to clarify them.